# Can perceptuo-motor skills outcomes predict future competition participation/drop-out and competition performance in youth table tennis players? A 9-year follow-up study

Irene R. Faber[1,2]*, Till Koopmann[1], Nicolette Schipper-van Veldhoven[2,3], Jos Twisk[4], Johan Pion[5,6]

1 Institute of Sport Science, Carl von Ossietzky University of Oldenburg, Oldenburg, Germany, 2 Research Centre Human Movement and Education, Windesheim University of Applied Sciences, Zwolle, The Netherlands, 3 Faculty of Behavioural, Management and Social Sciences, Faculty of Electrical Engineering, Mathematics and Computer Science, University of Twente, Enschede, The Netherlands, 4 Department of Epidemiology and Biostatistics, Amsterdam Public Health Research Institute, Amsterdam UMC Vrije Universiteit, Amsterdam, The Netherlands, 5 Institute for Studies in Sports and Exercise, HAN University of Applied Sciences, Nijmegen, The Netherlands, 6 Department of Movement and Sports Sciences, Faculty of Medicine and Health Sciences, Ghent University, Ghent, Belgium

* irene.faber@uol.de

**Data Availability Statement:** The data of this study cannot be made publicly available for ethical and

## Abstract

Tools that provide a fair estimate of young table tennis players' potential and their chances to succeed will support making decisions whether to commit to an extensive development program and the accompanying lifestyle. Consequently, this study included two research questions (RQ) to evaluate the capability of the Dutch perceptuo-motor skills assessment to predict competition participation/drop-out (RQ1) and competition performance (RQ2) in young table tennis players ($n = 39$; 7–11 years) using a tracking period of 9 years. The perceptuo-motor skills assessment consists of eight tests assessing gross motor function (i.e., sprint, agility, vertical jump) and ball control (i.e., speed while dribbling, aiming at target, ball skills, throwing a ball and eye-hand coordination). A Cox regression analysis demonstrated that a higher level of ball control was associated with a lower risk to drop-out from table tennis competition. The eye-hand coordination test appeared to be most suitable since it was the only test included in the multivariable Cox regression model (HR = .908; $p = .001$) (RQ1). Similarly, a multilevel regression analysis showed that a higher level of ball control was associated with a higher future competition performance. The eye-hand coordination and aiming at target tests were included in the multivariable multilevel model ($p < 0.05$; $R^2 = 36.4\%$) (RQ2). This evaluation demonstrates promising prospects for the perceptuo-motor skills assessment to be included in a talent development programme. Future studies are needed to obtain valid thresholds scores and clarify the predictive value in a larger sample of youth competition players.

legal reasons; the public availability would compromise confidentiality and/or participant privacy. The data contain potentially identifying athlete information. This restriction is imposed by the Netherlands Table Tennis Association. Data will only be available on request and can be sent to the Netherlands Table Tennis Association using the email address jong@tafeltennis.nl.

**Funding:** The author(s) received no specific funding for this work.

**Competing interests:** The authors have declared that no competing interests exist.

## Introduction

Programs for talent identification and development in children and adolescents have become an important pillar in modern sports [1, 2]. These so-called talent programs generally aim to support young athletes in discovering and developing their talents while encouraging as well as supporting the pursuit of excellence within (a certain) sport [3–5]. An important aspect of these programs is the identification of children with high potential for excellent adult performance (i.e., international level [6]) to provide the best opportunities for development already from a young age. This strategy is often employed to increase the effectiveness and/or efficiency of talent programs and to increase the chances for success by taking advantage of the most sensitive periods for learning [7, 8]. However, it is a challenge to adequately find and develop these 'diamonds in the rough' and prevent disappointing results (e.g., talent loss or drop-outs) [4, 5, 9]. As such, sports federations and clubs are searching for both established and innovative approaches to improve talent identification and development, to overcome setbacks and to create the best learning environment [10].

A recent scoping review by Baker and colleagues emphasized the necessity of prospective approaches including longitudinal tracking in talent programs and its paralleled research since talent in its essence is a time-constrained variable [11]. These approaches are considered to contribute to a better understanding of factors influencing the dynamics of individual pathways in the long run. This seems important especially for those sports that are characterized with a relatively long and intensive investment phase towards the elite level [12]. A fair estimate of an athlete's potential to reach the international level and the chances to stand on international podia will support the decision whether to commit to an extensive development program and the accompanying lifestyle. This information would serve everyone in the system, from the athletes, the coaches to the sport associations and other stakeholders. So-called talent programs are established approaches to talent promotion in a variety of sports [1]. One sport, in which the national associations typically aim to incorporate sufficient and functional talent identification and development programs is table tennis [13].

Table tennis is generally considered an early starting sport in which young players aiming for the elite levels already start at an age between 4 to 8. The age of peak performance, however, can rise up to 30 or even later, particularly for European players. As such, it can take up to 20 to 25 years of investment by a player and other stakeholders until the player reaches the elite level [13, 14]. Modern elite table tennis requires full dedication and commitment, often including difficult choices, such as leaving other sports, quit or change schools/education, transfer to another club, moving houses, adhere to a strict lifestyle and/or emigrate to another country [14]. A fair evaluation of a player's potential will help to judge whether all these sacrifices are worth the investment. Here, prospective observational research including the longitudinal tracking of youth table tennis players is expected to provide a deeper understanding of talent dynamics and the veracity of talent determinants. This knowledge can then support players, coaches, clubs and associations in designing (individual) pathways for all players [15].

Table tennis is a prime example of a technique-based sport [16]. Players aiming for the elite level need to develop outstanding technical skills including, among others, a proper body positioning and balance control, variable, flexible and fast footwork, and a fast-switching capability to adjust stroke techniques [16–18]. Moreover, technical skills are the basis for the ability to execute various tactical strategies (i.e., using an adequate solution to the given situational demands) [19, 20]. Therefore, coaches emphasize, besides a positive and safe learning climate, on the technical skill development (i.e., performing services and strokes under varying conditions) from the moment a youngster starts playing table tennis. Perceptuo-motor skills are fundamental for the development of outstanding sport-specific technical skills [19].

Consequently, the development towards elite level is highly dependent on a player's perceptuo-motor skills [14, 21, 22]. Accordingly, assessments of perceptuo-motor skills have become part of the evaluation of young table tennis players' potential [23–28] without down-playing that talent development in table tennis, as proposed by Baker et al. [29], still should be seen as a multidimensional (i.e. holistic), emergenic, dynamic and symbiotic process and that the accuracy of selection decisions is thus constrained by a range of factors [14, 29–31].

Based on this perspective, the Netherlands Table Tennis Association (NTTA) implemented a perceptuo-motor skills assessment with confirmed reproducibility and internal consistency as part of their talent program [28, 32]. This assessment was constructed by a team of experienced coaches with expertise in talent identification and embedded scientists on the basis of a task-analysis of elite table tennis. The assessment consisted of eight test items assessing the following essential perceptuo-motor skills for table tennis: eye–hand coordination, coordination of simultaneous foot and arm movements, combined gross and fine motor skills, agility, dynamic balance, bat and ball control, and high velocity in footwork [32]. It was hypothesized this assessment could contribute to the prediction of a player's competition participation as well as his/her competition performance. Players who have better perceptuo-motor skills are expected to be more motivated to start and remain participating in competition and perform better during competition. A first observational study including a 2.5 years follow-up in 48 young table tennis players (7–11 years) revealed that the assessment was not able to predict competition participation, but that that the outcomes of the perceptuo-motor skills assessment were significant predictors for future competition results ($R^2$ = 53%) [28]. However, as previously mentioned, the pathway to the elite level in table tennis is typically far longer. Thus, an evaluation of the perceptuo-motor skills assessment concerning its predictive validity for a longer period is essential for talent development purposes [5, 15]. For that reason, this 9-year follow-up study focuses on the following two research questions (RQ):

RQ1: Can the outcomes of the perceptuo-motor skills assessment predict competition participation/drop-out in young table tennis players?

RQ2: Can the outcomes of the perceptuo-motor skills assessment predict future competition performance in young table tennis players?

## Methods

### Ethical statement

This study and its informed consent procedure were approved by the ethical committee of the Medical Spectrum Twente (Medical School Twente, Institute for Applied Sciences, Enschede, the Netherlands; MTC/11069.oos 18-2-2011) in full compliance with the declaration of Helsinki. Written parental informed consent and players' consent were obtained prior to the testing.

### Study design

This explorative 9-year follow-up study used an observational prospective design to evaluate the predictive validity of a perceptuo-motor skills assessment regarding competition participation and competition performance outcomes of young table tennis players (age 7–11). The perceptuo-motor skills assessment was conducted in 2011 and 2012 and afterwards the players' competition participation and competition performance outcomes were monitored until January 2020. This period covered 18 or 16 consecutive competition periods of six months for the youth players tested in 2011 and 2012, respectively. A longer follow-up was considered inappropriate due to the COVID-19 pandemic allowing no (full) competitions to be played in 2020 and 2021.

## Players

The sample of young table tennis players ($\leq$ 11 years) consisted of the players that were recruited for the initial study [28] at the regional talent days of the eastern department of the NTTA in 2011 and 2012 and afterwards participated in at least one competition ($n$ = 39; nine players from the initial study never started to play competition within the tracking period). These players were selected and registered for the regional events by the coaches of their local clubs. The coaches were instructed to invite the youth players with the highest potential for regional and/or national elite table tennis. The eastern department is one of eight regional competition departments connected to the NTTA. Its total population of young players was estimated to be between 100 and 120 players per year at that time.

## Perceptuo-motor skills assessment

The perceptuo-motor skills assessment consists of three tests assessing gross motor function (i.e., sprint, agility, vertical jump) and five tests assessing ball control (i.e., speed while dribbling, aiming at target, ball skills, throwing a ball and eye-hand coordination). 'Sprint' included a pyramid-shape circuit in which players need to gather and return five table tennis balls one by one as fast as possible from five different baskets starting at the basis of the pyramid-shaped circuit (measured in s). For 'agility', players needed to get through a circuit, including climbing over a gymnastics' cabinet and under and over a low hurdle as fast as possible (measured in s). At 'vertical jump' players were instructed to jump as high as possible and touch the wall at the highest point possible. The difference between the jumping height and standing height with one arm up along the wall was measured in centimetres. 'Speed while dribbling' used a zigzag circuit in which the players needed to move sideways as fast as possible while dribbling with a basketball using one hand (measured in s). At 'aiming at target' players needed to hit a round target (Ø 60 cm) on the floor at 2.5-meter distance with a table tennis ball using a standard bat with their preferred hand (measured as points made). 'Balls skills' also required hitting a round target on the floor (Ø 75 cm), but here players needed to throw a table tennis ball with their preferred hand via a vertical table tennis table from two different positions, 1 and 2 meter distance away from the target (measured as points made). At 'throwing a ball', the players threw a table tennis ball as far away as possible with their preferred hand (measure in m). In the 'eye-hand coordination' test players were instructed to throw a ball at a vertical table tennis table at 1 meter distance with one hand and to catch the ball correctly with the other hand as frequently as possible in 30 seconds (measured as points made). The complete test protocol of the assessment is available online [28]. Previous evaluations of the perceptuo-motor skills assessment demonstrated fair to good reproducibility with regard to the level of test items (ICC = 0.81–0.85; $p < 0.001$; CV = 3–19%) and the total score (ICC = 0.91; $p < 0.001$; SDD = 98 points; CV = 7%). The internal consistency of all test items is satisfactory (Cronbach's alpha = 0.82), and the validity (i.e., the association between the assessment's outcomes and current competition performance and the assessment's discriminative ability between performance levels) is considered moderate to good [28, 32].

All children were tested under similar conditions as part of the regional talent days after conducting a warm-up. Total testing time for the perceptuo-motor tests was approximately 20 minutes for each child spread over three sessions. Test leaders were physical therapy students or table tennis coaches who were familiarized with the use of the test protocol. Instruction and feedback were given during a practical training by an expert coach of the NTTA. The youth players' characteristics for height, weight and current training hours per week as well as the control variables of sex (m/f) and age (years) were extracted from the register forms.

## Competition participation/drop-out and competition performance

Youth players included in this study participated in the official competitions of the NTTA including both team competitions and individual tournaments. One calendar year includes two competition periods of six months with ten team matches and approximately three to five individual tournaments. Competition participation/drop-out and competition performance data were extracted from the open archives of the NTTA (https://www.nttb-ranglijsten.nl/ranking.php). Competition participation (RQ1) was considered nominal data (yes/no) per competition period. Consequently, it was determined how many competitions periods a player participated within the follow-up period and if a player dropped-out from competition (yes/no) and if so at what age (years). Competition rating scores (ratio data) indicated the player's competition performance (RQ2); the higher the rating score the better the player's table tennis competition performance. For example, players from the lowest regional youth leagues have a score approximately between 0 to 200 points and the players of the highest national youth competition a score approximately between 1400 to 1800 point. The calculation of the competition rating score (https://www.nttb-ranglijsten.nl/elo300.php) is based on the official NTTA's national and regional competitions and can be converted to international standards. It allows to compare performances levels between players (youth and adult players, male and female players) who participate in any of the regional and national competition leagues and tournaments [28]. As such, a player's table tennis performance is ranked within a certain competition period compared to all players competing in the same and other leagues.

## Statistical analysis

Both IBM SPSS Statistics 26 (IBM Corp. Released 2019. IBM SPSS Statistics for Windows, Version 26.0. Armonk, NY: IBM Corp) and STATA 17 (StataCorp. 2021. Stata Statistical Software: Release 17. College Station, TX: StataCorp LLC) were used for the statistical analyses. The normality of the competition performance outcome was evaluated by comparing mean and median and by visual inspection. Descriptive statistics of the raw test item scores are presented for the total sample and for boys and girls separately. Then firstly, a Cox regression analysis was used to examine if perceptual motor skills (independent variables) predicted competition participation/drop-out (dependent variable) within the follow-up period of this study (RQ1). Secondly, a multilevel regression analysis was conducted to explore the predictive value of the perceptuo-motor skills outcomes (independent variables) for the longitudinal competition performance outcomes (dependent variable) of the 9-year follow-up period (RQ2). For this analyses, the repeated observations (level 1) were correlated within the subjects (level 2). For both the Cox regression analysis and the multilevel regression analysis, univariable and multivariable models were created. The univariable models provide insight into the predictive value of a single test item, while the multivariable analyses evaluate the strength of and relationship between the items when used as part of the perceptuo-motor skills assessment. For the final multivariable models, a backward selection procedure was used with a cut-off p-value of 0.05; so all variables in the final multivariable model were significantly related to the outcome. Besides that, for the final multivariable multilevel model, the explained variance was reported. Based on the results of previous studies, sex and test age were included as covariates in all analyses. The significance level alpha was set at 0.05 for all analyses.

## Results

### Sample characteristics and descriptive results

A total of 39 young table tennis players (21 boys and 18 girls, mean age = 9.4, SD = 1.06 years) participated in this study; 21 from the regional talent day in 2011 (10 boys and 11 girls) and 18

**Table 1. Descriptive statistics and Cox regression analysis of dropout rates examining the perceptuo-motor skills assessment.**

| Univariable | Total group | Boys | Girls | HR | p | 95% confidence interval |
|---|---|---|---|---|---|---|
| | M (SD) | M (SD) | M(SD) | | | |
| | n = 39 | n = 21 | n = 18 | | | |
| Sprint (s)[a] | 34.8 (3.6) | 33.7 (6.7) | 36.2 (3.2) | 1.115 | .069 | .992–1.254 |
| Agility (s)[a] | 24.7 (4.6) | 22.9 (5.5) | 26.8 (5.2) | .996 | .926 | .909–1.091 |
| Vertical jump (cm) | 30.7 (5.9) | 30.7 (6.2) | 30.6 (5.7) | 1.067 | .130 | .981–1.160 |
| Speed while dribbling (s)[a,1,2] | 24.1 (6.6) | 21.5 (4.7) | 27.1 (7.4) | 1.076* | .042 | 1.003–1.154 |
| Aiming at target (points) | 22.9 (10.3) | 24.5 (11.5) | 21.0 (8.6) | .963 | .054 | .927–1.001 |
| Ball skills (points)[1,2] | 19.3 (6.0) | 21.7 (6.3) | 16.4 (5.4) | .910* | .038 | .832 - .995 |
| Throwing a ball (m)[2] | 9.4 (1.5) | 10.1 (1.4) | 8.5 (1.2) | .753 | .063 | .558–1.016 |
| Eye-hand coordination (points)[1] | 13.2 (7.5) | 14.5 (7.8) | 11.6 (7.0) | .908* | .001 | .858 - .960 |
| **Multivariable** | | | | HR | p | 95% confidence interval |
| Eye-hand coordination (points) | | | | .908* | .001 | .858 - .960 |

HR: Hazard Ratio

*$p < 0.05$; [a]Lower values indicate better performance.

Test age and sex were included as covariates in all models

[1]test age and

[2]sex significant covariates ($p < 0.05$).

from the regional talent day in 2012 (11 boys and 7 girls). This number amounts to approximately 20% of the players available in this age category in the eastern department per year. All raw test scores of the motor skills assessment were evaluated as normally distributed; means and medians were similar and the range around the mean followed a normal distribution. There were no missing data. Descriptive results of the perceptuo-motor skills assessment are presented in Table 1.

## Predicting competition participation/drop-out (RQ1)

The univariable Cox regression analyses used to evaluate whether the perceptuo-motor skills assessment (i.e., test results) predicts competition drop-out demonstrated three significant predictors: speed while dribbling (HR = 1.076; $p = .042$), ball skills (HR = .910; $p = .038$) and eye-hand coordination (HR = .908; $p = .001$) (Table 1). Test age and sex were significant covariates within the univariable models of speed while dribbling and ball skills, while sex was also a significant covariate within the univariable model of throwing a ball ($p < 0.05$). The multivariable analysis with backwards procedure showed that the final multivariable model consisted of only one test item: eye-hand coordination (HR = .908; $p = .001$). This means that per ball a player catches, he/she has on average .908 times the chance to drop-out over time. That is, a higher score for eye-hand coordination appears to have a preventive value for dropping out of the sport.

## Predicting competition performance (RQ2)

The results of the multilevel analyses are presented in Table 2. Six of the eight test items significantly predicted the longitudinal competition outcomes in univariable models; sprint (B = -45.450; $p = .001$), speed while dribbling (B = -27.152; $p = .003$), aiming at target (B = -17.977; $p < .001$), ball skills (B = 26.311; $p = .007$), throwing a ball (B = 131.080; $p = .003$), and eye-hand coordination (B = 34.260; $p < .001$). Test age was a significant covariate within the univariable models of speed while dribbling, ball skills, throwing a ball and eye-hand coordination

**Table 2. Multilevel analysis for predictive value of the perceptuo-motor skills assessment on competition performance.**

| Univariable | B | p | 95% confidence interval | $RR^2$ (%) |
|---|---|---|---|---|
| Sprint (s) | -45.450* | .001 | (-72.091 - -18.809) | 15.1 |
| Agility (s) | -12.993 | .283 | (-36.725–10.739) | 2.1 |
| Vertical jump (cm) | -3.248 | .747 | (-22.943–16.446) | 0.0 |
| Speed while dribbling (s)[1] | -27.152* | .003 | (-44.931 - -9.373) | 12.7 |
| Aiming at target (points) | 17.977* | < .001 | (9.536–26.417) | 19.3 |
| Ball skills (points)[1] | 26.311* | .007 | (7.104–45.519) | 9.2 |
| Throwing a ball (m)[1] | 131.080* | .003 | (45.501–216.659) | 11.8 |
| Eye-hand coordination (points)[1] | 34.260* | < .001 | (23.489–45.032) | 31.3 |
| **Multivariable**[1] | B | p | 95% confidence interval | $R^2$ (%) |
| Aiming at target (points) | 10.189* | .005 | (3.035–17.340) | 36.4 |
| Eye-hand coordination (points) | 27.769* | < .001 | (16.984–38.554) | |

B:regression coefficient

*$p < 0.05$.

Test age and sex were included as covariates in all models

[1]test age and

[2]sex significant covariates ($p <0.05$).

($p < 0.05$). The final multivariable model had an explained variance of 36.4% and included the test items aiming at target (B = 10.189; $p$ = .005) and eye-hand coordination (B = 27.769; $p < .001$). Test age was also a significant covariate in this multivariable model ($p < 0.05$).

## Discussion

This study focused on the capacity of a perceptuo-motor skills assessment to predict future competition participation/drop-out and competition performance in young table tennis players. The results of this 9-year follow-up study demonstrated that a higher level of ball control was associated with a lower risk to drop-out from table tennis; three tests for ball control (i.e., speed while dribbling, ball skills, and eye-hand coordination) significantly predicted players' participation/drop-out. The eye-hand coordination test appeared to be most suitable since it was the only test included in the multivariable model. Similarly, a higher level of ball control also appears to be associated with a higher future competition performance; all five ball control tests (i.e., speed while dribbling, aiming at target, ball skills, throwing a ball, and eye-hand coordination) significantly predicted the longitudinal competition scores. Although the sprint test was found to be a significant predictor as well, only two of the ball control tests were included in the multivariable model (i.e., eye-hand coordination and aiming at target). These results are in line with previous studies showing that perceptuo-motor skills play an important role in the development of young athletes' competition performance in sports consisting of complex motor tasks, and that specific tests to assess the underlying skills inherent to a particular sport are appropriate for estimating the potential regarding the perceptuo-motor domain [33–36].

The explained variance of the multilevel model to predict competition performance (36.4%) was a bit lower compared to the model in the previous study (53%) [28]. This is likely due to the longer tracking-period in the current study which comes with an increasing influence of other performance-determining factors including mental aspects (e.g., motivation, self-efficacy, volition, and self-esteem), contextual factors (e.g., training facilities and parental support) and the learning environment (e.g. positive and safe sport climate, team support,

coaching style) [37, 38]. These other factors might become quite important determinants during puberty/adolescence which covers a great part of the investment phase in table tennis [12]. Moreover, since table tennis performance characteristics are multidimensional, a weaker performance on the one aspect could be compensated by other traits (i.e., compensation phenomenon) [5]. Still, the perceptuo-motor skills assessment and specifically the ball control tests seem to contribute to make fair decisions during the selection procedures and, therefore, seem to be suitable as a part of multidimensional profiling of young table tennis players.

When tests are included as part of the selection process, it is helpful to establish threshold scores that are associated with continued participation (i.e., survival) and the future competition performance [36]. This would support the estimation of a player's physical and technical potential and can provide guidelines for a player's individual pathway. However, more data are needed to be able to calculate threshold scores specifically valid for table tennis youth players from different age groups and for boys and girls separately. The predictive value/weight and threshold scores of the test items within the assessment might differ between sexes due to, among other things, differences in physical appearances and personal preferences between boys and girls that influence the development of a player's playing style.

Additionally, some criticism should be pointed out regarding the current 'talent system' in which the assessment was used. If you consider the pathway of a young player within this system, there are mainly two critical issues. First, it is known that in 2011 and 2012 there were approximately 100–120 registered children within the U11 age category. Only 20–25% of this group participated at the regional talent day. This means that 75–80% of the children were not tested and had therefore no opportunity to be included in the selection for the consecutive national talent day. This early 'deselection' can be based on the judgement of their skills by their coaches but could also be due to the lack of response by their club. Second, some of the children never started to play competitions. This again can be due to individual constraints (e.g., motivation, self-esteem), but in some cases may be due to environmental constraints (e.g., no team available, club culture). Both issues ensure a considerable loss within the talent pool and cause a weakness in the talent system. Efforts to overcome these issues are likely to contribute to increases in the effectiveness and efficiency of the talent program.

Limitations of this study need to be acknowledged. First, as previously mentioned, this study included only a small sample recruited during the regional talent day in the eastern department and pre-selected by club coaches. The generalisation of the findings to the players participating at the national talent day, including selected players from other regions, is expected to be valid given the sufficient variety in the test scores and competition rating in the 'upper part' of the sample. However, this should be verified in a larger sample including players from the national talent day since the assessment might lose its strength in a more homogenous sample regarding perceptuo-motor skills. Second, to optimize the power of this study, the survival and multilevel analyses were conducted using the total sample. The outcomes of both the boys and girls within the test age spanning from 7 to 11 years were analysed altogether. Consequently, the analyses included different developmental stages in which the associations were calculated, especially when considering influences of differences in growth and maturation [39]. Although sex and test age were included as covariates, it is recommended for future research to split the analyses for boys and girls as the weight of the test items might differ between sexes (i.e., interaction effect), to reduce the age span for prediction models and to take into account growth and maturation.

In conclusion, this study reinforces the promising prospects regarding the predictive validity of the perceptuo-motor skills assessment as part of a multidimensional assessment as a helpful tool for coaches regarding talent purposes in table tennis. This assessment can contribute by objectifying a young player's potential regarding the perceptuo-motor domain and for

that reason support decisions/selections. It has to be emphasized, however, that talent development is a multidimensional process [15, 29, 40] and that the accuracy of selection decisions is influenced by a range of factors [14, 29–31]. Moreover, to interpret an individual player's test scores, it is important that thresholds will be based on a larger dataset and that the player's sex, test age, training experience, growth and maturity level are taken into account [36, 39]. Furthermore, talent development programs do not intend to limit children's freedom of choice to practice a particular sport. Coaches should also be aware of the potential risks of early specialization and selection (e.g., injuries, mental exhaustion and drop-outs) [41, 42] and should always create a safe and positive learning climate [43]. This is crucial to both preventing early drop-out and stimulating development. Finally, the perceptuo-motor skills assessment is only intended to identify those children excelling in this essential performance aspects in table tennis.

## Acknowledgments

We acknowledge the Netherlands Table Tennis Association for the provision of the data and Wilke Epkes for his help completing the dataset.

## Author Contributions

**Conceptualization:** Irene R. Faber, Till Koopmann, Nicolette Schipper-van Veldhoven, Jos Twisk, Johan Pion.

**Data curation:** Irene R. Faber.

**Formal analysis:** Irene R. Faber, Jos Twisk.

**Investigation:** Irene R. Faber.

**Methodology:** Irene R. Faber, Jos Twisk, Johan Pion.

**Project administration:** Irene R. Faber.

**Software:** Irene R. Faber.

**Supervision:** Irene R. Faber.

**Validation:** Irene R. Faber.

**Writing – original draft:** Irene R. Faber.

**Writing – review & editing:** Till Koopmann, Nicolette Schipper-van Veldhoven, Jos Twisk, Johan Pion.

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
