## [Decision Letter · Decision Letter 0]

13 Oct 2022

PONE-D-22-23471Can perceptuo-motor skills assessment outcomes predict future competition participation/drop-out and performance level in youth table tennis players? An explorative 9-year follow-up studyPLOS ONE

Dear Dr. Faber,

Thank you for submitting your manuscript to PLOS ONE. After careful consideration, we feel that it has merit but does not fully meet PLOS ONE’s publication criteria as it currently stands. Therefore, we invite you to submit a revised version of the manuscript that addresses the points raised during the review process.

ACADEMIC EDITOR:Dear authors,

The authors presented an interesting work. However, there are some concerns that need attention before a further considerations.

Please consider comments made by both reviewers. There some issues that need better clarity to understand all procedures applied in the manuscript and also to provide a proper rationale of the whole design and aims of the paper. Thus, I suggest a round major revisions at this stage.

Thank you

We look forward to receiving your revised manuscript.

Kind regards,

Rafael Franco Soares Oliveira

Academic Editor

PLOS ONE

Journal Requirements:

Additional Editor Comments:

Dear authors,

The authors presented an interesting work. However, there are some concerns that need attention before a further considerations.

Please consider comments made by both reviewers. There some issues that need better clarity to understand all procedures applied in the manuscript and also to provide a proper rationale of the whole design and aims of the paper. Thus, I suggest a round major revisions at this stage.

Thank you

Reviewers' comments:

Reviewer's Responses to Questions

**Comments to the Author**

1. Is the manuscript technically sound, and do the data support the conclusions?

Reviewer #1: Yes

Reviewer #2: Yes

2. Has the statistical analysis been performed appropriately and rigorously? 

Reviewer #1: Yes

Reviewer #2: N/A

3. Have the authors made all data underlying the findings in their manuscript fully available?

Reviewer #1: Yes

Reviewer #2: Yes

4. Is the manuscript presented in an intelligible fashion and written in standard English?

Reviewer #1: Yes

Reviewer #2: Yes

5. Review Comments to the Author

Reviewer #1: Can perceptuo-motor skills assessment outcomes predict future competition participation/drop-out and performance level in youth table tennis players? An explorative 9-year follow-up study

First of all, the reviewer would like to thank the authors for their work and efforts in trying to improve sports science knowledge. The authors are commended on their efforts thus far. The article is an interesting approach to assessment outcomes predict future competition participation/drop-out and performance level in youth table tennis players. The study is well designed and well-written, with a great original article evaluating the usefulness of the topic.

Abstract

Line 51: eye-hand or eye hand please check and fix the throughout the manuscript

Introduction

This section is well designed and well-written. However, here is the alternative sentence for the research questions.

Line 117-122: The purposes of this study were (a) can the outcomes of the perceptuo-motor skills assessment predict competition participation/drop-out in young table tennis players, (b) can the outcomes of the perceptuo-motor skills assessment predict future performance level in young table tennis players (c) relationships between psychophysiological responses and locomotor demands.

Methods section

Line 156: no need this info https://doi.org/10.1371/journal.pone.0149037.s001. Please extract it.

What about maturation. If you do not measure it, please add limitations.

Results section

Results and tables are well shown

Discussion section

Overall the discussion is well-written and incorporates relevant literature.

References

References are well selected by the authors

Figures and Tables

This section is well designed and well-shown.

Reviewer #2: Reviewer Comments:

Can perceptuo-motor skills assessment outcomes predict future competition

participation/drop-out and performance level in youth table tennis players? An explorative 9-year follow-up study: PONE-D-22-23471

GENERAL COMMENTS:

Thank you for your contribution to PLoS One. Overall, the paper which presents a study analysing the (prognostic) validity of motor performance in young table tennis players falls within the scope of the journal and should be of interest to the readership. Strengths of the study mainly relate to the investigation of important talent predictors at a young age (7-11) years for long-term success while utilizing a 9-year prognostic period in a sport where talent research is still scarce. However, in its current form the manuscript also presents a range of concerns in regard to its content (rationale and theoretical background of assessed predictors, definition of criterion variables, discussion of results) and presentation (particularly description and clarity of utilized methodology).

With regards to the content of the paper, some aspects require improvements.

1) First, I would ask you to elaborate on the explanation of the assessed constructs in the introduction. It should be more highlighted that the constructs you are assessing cover only some small pieces of the multifaceted talent characteristics needed for an excellent young table tennis players. Maybe you could just use a theoretical model of talent development (like the one of Gagné) or a heuristic model of talent predictors (although I am currently only aware of a soccer-specific one by Williams and Reilly, 2000) in order to show the complexity of talent in the sport of table tennis and then present the area you want to focus on. Why exactly those motor skills were chosen? Please provide rationale in the introduction.

2) Much more important are my concerns with regard the separation of the two research questions (RQ) and the utilized criterion variables for those. to purpose 3 (validity of physical tests for in-game soccer performance). It is not entirely clear to me, why exactly the two RQs were differentiated and chosen. What is RQ1 for, what is the aim of RQ2 – both are evaluating the long-term prognostic validity, by focussing potentially different things – participation and performance level. If this is the reason, this should be explained in more detail. Potentially the lacking clarity is also a result of some inconsistencies in the use of the chosen criterion: while for RQ2 future performance level is used for instance in the title, sometimes also competition performance or competition performance level are used. I would recommend being consistent here. By the way: Please think about labelling your two RQs as RQ1 and RQ2 and please use that structure (including subheadings) also for the methods and results part. This could facilitate readability.

3) Further, there are some severe issues with the description and the clarity of the utilized statistical analyses. Although I generally agree with the choice of them, a final decision of the adequate use is not possible due to some missing information on the analyses and results you present. Please exactly describe and state the procedure you were following in each of your research question regarding the analyses: why and how did you exactly perform Cox regression analysis in the univariate, how in the multivariate approach. Why is the use of Cox regression (what I appreciate) useful and appropriate for your setting. Just to give you an example: It ist not clear by the means, what the survival curve in Figure 1 exactly display/how it was computed. Is it a result of the multivariate Cox regression or is it “just” the Kaplan-Meier Curve with survival rates and hazard ratios computed based on “drop-outs” in specific time intervals. Please elaborate, otherwise it is not possible to follow your results.

4) The same applies for RQ2 [performance level]: which multilevel analyses did you use? Multiple Regression? What are your levels (are there some?) and why this is intended. If so, where are the random effect that were estimated /why were they not?

5) Furthermore, with regard to the utilized criterion variables, please clarify how exactly performance level was measured. There is some information given about a score, but this is not detailed enough. Please elaborate and please also refer to the former comment regarding consistency in labelling/defining your main outcomes/criterions in the former comment.

6) Moreover, I really struggle with your approach to aggregate performance outcomes of boys and girls. I know, that this was followed due to the rather low sample size and I appreciate reading about this short-coming in the discussion (and considering gender at least as a covariate). However, I think you should enlarge this part of the discussion by giving some information on potential similarities or dissimilarities of talent pathways/ differences in performance/ as to whether the utilized performance scores are comparable among gender. Apart from that, I would really like to see the results of the utilized covariates e.g., gender also in your tables. There are only some hints on covariates without providing the results for them.

SPECIFIC COMMENTS

Title

• “assessment outcomes” sounds a bit confusing- only assessments or only outcomes or just diagnostics?

• Why explorative? Please add a rational in the text. Is this important for the title?

Abstract

• L.44: decision-making: think about replacing by making decisions regarding selection/promotion (or similar). Decision-making bis connoted with making in-game decisions and therefore could be misleading here.

• L.46: competition performance vs. future performance level? See general comment

• L.54: add regression?

Introduction

• L.66: please define excellent adult performance

• L.72: provide reference

• L.75: I agree that longitudinal approaches are needed. But, please be aware that your approach is prospective, but in my point of view not longitudinal – in the sense that you assessed predictors only once. Please precise.

• L.78 and L.90: what do you mean by fair estimate /evaluation of potential. This should be explained (also it might be implicitly clear)

• L.81: maybe add that talent development programs are established approaches of talent promotion in a variety of sports with some examples, the focus on table tennis)

• L117: explain, why the study is explorative

Methods

• l.144 provide distribution regarding gender and age

• Is there a rule, in which ages the players are nominated for the squads? If I am right, players were between 7 and 11 years old, which seems to be a rather large range for initial talent identification events (if the events were of such type?)

• L.157pp: please double check the presentation of the statistical values. Should read e.g., ICC = 0.91. Please also check formatting (e.g., n should be in italics, see for instance l.144)

• L166-168: sentence seems not to be complete.

• L.170pp: Please explain which “levels” were exactly used (or whether level is the right term for the score that was used). See also general comment.

• L.187: was there also significance testing for normality?

Results:

• L.215: lower values are better performances in speed measures, right? Please note on that, as this is important for the interpretation of the HRs.

• By the way: although I like the idea of referring to the diagnostics elsewhere, it would be helpful to at least provide the units /general shape of the utilized tests) in order to better understand interpretations in this part.

• “best fitting model”: please provide measures for that (and describe the process how it was found in more detail)

• L.226pp: Please state what was used as independent and dependent variables in the regressions (already in the methods section) and add this information also in Table 2.

Discussion

L.250: decimal point instead of comma

L.261: commas before and after “therefore”?

l.290: see general comment on combined approach regarding gender

L.195pp: In line with my comment regarding the introduction of the utilized and investigated measure, I would recommend to extend this paragraph by giving more information why the study “only” focused on a small part of the complex construct of talent and to highlight that this small piece is more a support for coaches and not a replacement (?).

Figure 1:

• See general comment. I would recommend deleting ages 0-7 years (not included in the dtudy, and players were potentially not in the system yet. Please also provide Confidence intervals for the survivals (in order to get insight in potential problems due to lower sample sizes for later drop outs). Please add also a label for the columns with n=21 and n=18 (presumably the 2011 and 2012 assessments separately?)

6. PLOS authors have the option to publish the peer review history of their article (what does this mean?). If published, this will include your full peer review and any attached files.

Reviewer #1: No

Reviewer #2: No

---

## [Author Response · Author response to Decision Letter 0]

30 Oct 2022

Point-by-point response

Journal Requirements:

Response

We have addressed all points to the best of our knowledge, and have revised the manuscript accordingly. Since these revisions were not substantive changes, they are not indicated as tracked changes within the manuscript to maintain readability. 

We need to change our data availability statement. The data of this study cannot be made publicly available for ethical and legal reasons; the public availability would compromise confidentiality and/or participant privacy. The data contains potentially identifying athlete information. This restriction is imposed by the Netherlands Table Tennis Association. Data will only be available on request and can be sent to the Netherlands Table Tennis Association using the email address jong@tafeltennis.nl.

The ethics statement is now only stated within the methods section. 

Reviewers' comments:

Reviewer #1: Can perceptuo-motor skills assessment outcomes predict future competition participation/drop-out and performance level in youth table tennis players? An explorative 9-year follow-up study

First of all, the reviewer would like to thank the authors for their work and efforts in trying to improve sports science knowledge. The authors are commended on their efforts thus far. The article is an interesting approach to assessment outcomes predict future competition participation/drop-out and performance level in youth table tennis players. The study is well designed and well-written, with a great original article evaluating the usefulness of the topic.

Response

Thank you for your positive response regarding our paper and the feedback that helps to improve its content and readability. 

Abstract

Line 51: eye-hand or eye hand please check and fix the throughout the manuscript

 

Response

Thank you for you careful reading. We have changed eye hand to eye-hand throughout the manuscript. 

Introduction

This section is well designed and well-written. However, here is the alternative sentence for the research questions.

Line 117-122: The purposes of this study were (a) can the outcomes of the perceptuo-motor skills assessment predict competition participation/drop-out in young table tennis players, (b) can the outcomes of the perceptuo-motor skills assessment predict future performance level in young table tennis players (c) relationships between psychophysiological responses and locomotor demands.

Response

Thank you for your suggestion. However, we are a little bit puzzled by the alternative, especially regarding the inclusion of question (c) relationships between psychophysiological responses and locomotor demands. We believe that this was beyond the scope of our article. Nevertheless, we tried to simplify the research question, please see line 107-112.

Methods section

Line 156: no need this info https://doi.org/10.1371/journal.pone.0149037.s001. Please extract it.

What about maturation. If you do not measure it, please add limitations.

Response

We have deleted the link and added the maturation as a topic to the limitation paragraph (line 317-319) as proposed.

Results section: Results and tables are well shown

Discussion section: Overall the discussion is well-written and incorporates relevant literature.

References: References are well selected by the authors

Figures and Tables: This section is well designed and well-shown.

Response

Thank you for these compliments and again for your feedback. 

Reviewer #2: Reviewer Comments:

Can perceptuo-motor skills assessment outcomes predict future competition

participation/drop-out and performance level in youth table tennis players? An explorative 9-year follow-up study: PONE-D-22-23471

GENERAL COMMENTS:

Thank you for your contribution to PLoS One. Overall, the paper which presents a study analysing the (prognostic) validity of motor performance in young table tennis players falls within the scope of the journal and should be of interest to the readership. Strengths of the study mainly relate to the investigation of important talent predictors at a young age (7-11) years for long-term success while utilizing a 9-year prognostic period in a sport where talent research is still scarce. However, in its current form the manuscript also presents a range of concerns in regard to its content (rationale and theoretical background of assessed predictors, definition of criterion variables, discussion of results) and presentation (particularly description and clarity of utilized methodology).

 

Response

Thank you for your positive response and critical feedback that helps us to improve our paper. 

With regards to the content of the paper, some aspects require improvements.

1) First, I would ask you to elaborate on the explanation of the assessed constructs in the introduction. It should be more highlighted that the constructs you are assessing cover only some small pieces of the multifaceted talent characteristics needed for an excellent young table tennis players. Maybe you could just use a theoretical model of talent development (like the one of Gagné) or a heuristic model of talent predictors (although I am currently only aware of a soccer-specific one by Williams and Reilly, 2000) in order to show the complexity of talent in the sport of table tennis and then present the area you want to focus on. Why exactly those motor skills were chosen? Please provide rationale in the introduction.

Response

Thank you for your suggestions. Indeed, our starting point is that talent development is a multidimensional process in which many factors play a role. We now emphasize on that in the introduction in line 86-90 by including the conceptualization of talent as proposed by prof. Baker and colleagues. Additionally, psychomotor skills are important elements of performance in the sports domain and especially in technique-based sports like table tennis. We have added a more detailed rational about the assessed constructs (line 93-98). 

Added references: 

• Baker J, Wattie N, Schorer J. A proposed conceptualization of talent in sport: The first step in a long and winding road. Psychology of Sport and Exercise. 2019;43:27-33.

• Faber IR, Bustin PM, Oosterveld FG, Elferink-Gemser MT, Nijhuis-Van der Sanden MW. Assessing personal talent determinants in young racquet sport players: a systematic review. Journal of sports sciences. 2016;34(5):395-410.

• Faber IR, Nijhuis-Van Der Sanden MW, Elferink-Gemser MT, Oosterveld FG. The Dutch motor skills assessment as tool for talent development in table tennis: a reproducibility and validity study. Journal of sports sciences. 2015;33(11):1149-58.

2) Much more important are my concerns with regard the separation of the two research questions (RQ) and the utilized criterion variables for those. to purpose 3 (validity of physical tests for in-game soccer performance). It is not entirely clear to me, why exactly the two RQs were differentiated and chosen. What is RQ1 for, what is the aim of RQ2 – both are evaluating the long-term prognostic validity, by focussing potentially different things – participation and performance level. If this is the reason, this should be explained in more detail. Potentially the lacking clarity is also a result of some inconsistencies in the use of the chosen criterion: while for RQ2 future performance level is used for instance in the title, sometimes also competition performance or competition performance level are used. I would recommend being consistent here. By the way: Please think about labelling your two RQs as RQ1 and RQ2 and please use that structure (including subheadings) also for the methods and results part. This could facilitate readability.

Response

Indeed, what you describe is exactly what we aimed for. We wanted to evaluate the prognostic value for both competition participation and competition performance. We provided more detailed information on this at the end of the introduction section (line 98-101). We also now use RQ1 and RQ2 for the research questions as you suggested (line 107-112) and the structure of the results section. Also, we checked the consistency of terms regarding competition performance throughout the paper. 

3) Further, there are some severe issues with the description and the clarity of the utilized statistical analyses. Although I generally agree with the choice of them, a final decision of the adequate use is not possible due to some missing information on the analyses and results you present. Please exactly describe and state the procedure you were following in each of your research question regarding the analyses: why and how did you exactly perform Cox regression analysis in the univariate, how in the multivariate approach. Why is the use of Cox regression (what I appreciate) useful and appropriate for your setting. Just to give you an example: It ist not clear by the means, what the survival curve in Figure 1 exactly display/how it was computed. Is it a result of the multivariate Cox regression or is it “just” the Kaplan-Meier Curve with survival rates and hazard ratios computed based on “drop-outs” in specific time intervals. Please elaborate, otherwise it is not possible to follow your results.

Response

As the reviewer correctly noticed, Figure 1 is a Kaplan-Meier curve which shows the survival function (i.e., probability to remain in competition) against time (i.e., the age of a player). This information is now added to the methods section (line 202-203) and results section (line 233-234).The x-axis title is changed for clarity into age (years) and for the y-axis to survival probability. The Cox regression analyses are first conducted for each item separately (univariable), to highlight the individual interest of each of the measured variables. After that also multivariable analyses were conducted. 

4) The same applies for RQ2 [performance level]: which multilevel analyses did you use? Multiple Regression? What are your levels (are there some?) and why this is intended. If so, where are the random effect that were estimated /why were they not?

Response

Additional information is provided in the statistical paragraph of the methods section (line 206-215) to better explain the analyses that were conducted. 

5) Furthermore, with regard to the utilized criterion variables, please clarify how exactly performance level was measured. There is some information given about a score, but this is not detailed enough. Please elaborate and please also refer to the former comment regarding consistency in labelling/defining your main outcomes/criterions in the former comment.

Response

As proposed, we added information about the competition rating score in line 185-194. Hopefully, this provides enough information. Please let us know, whether specific information is still missing. 

6) Moreover, I really struggle with your approach to aggregate performance outcomes of boys and girls. I know, that this was followed due to the rather low sample size and I appreciate reading about this short-coming in the discussion (and considering gender at least as a covariate). However, I think you should enlarge this part of the discussion by giving some information on potential similarities or dissimilarities of talent pathways/ differences in performance/ as to whether the utilized performance scores are comparable among gender. Apart from that, I would really like to see the results of the utilized covariates e.g., gender also in your tables. There are only some hints on covariates without providing the results for them.

Response

Although we indeed agree that an analysis including sex would be the better solution, this is considered inappropriate while including a small dataset. Therefore, we only corrected for sex by including sex as a covariate. It is quite common not to report the coefficients and p-values of the covariates in a regression model. The coefficients of the covariates are not part of the research question and therefore, they don't provide relevant information. That's the reason why the coefficient for age and sex were not reported. Anyway, if you still want to see them, of course, we can provide, but again, we don't believe that this information is very relevant. In addition to this, we emphasize in the discussion part on a future approach in which sex is taken into account (line 291-293; 312-319; 326-328). 

SPECIFIC COMMENTS

Title

• “assessment outcomes” sounds a bit confusing- only assessments or only outcomes or just diagnostics?

• Why explorative? Please add a rational in the text. Is this important for the title?

Abstract

• L.44: decision-making: think about replacing by making decisions regarding selection/promotion (or similar). Decision-making is connoted with making in-game decisions and therefore could be misleading here.

• L.46: competition performance vs. future performance level? See general comment

• L.54: add regression?

Response

We have made changed to the manuscript as proposed: 

- Assessment is deleted. 

- Explorative Is deleted; this was mainly due to the small sample, but seems unnecessary.

- Decision making is changed into making decisions or similar wording throughout the paper to avoid misunderstandings. 

- Competition performance is used throughout the document. 

- Regression is added to the description.

Introduction

• L.66: please define excellent adult performance

• L.72: provide reference

• L.75: I agree that longitudinal approaches are needed. But, please be aware that your approach is prospective, but in my point of view not longitudinal – in the sense that you assessed predictors only once. Please precise.

• L.78 and L.90: what do you mean by fair estimate /evaluation of potential. This should be explained (also it might be implicitly clear)

• L.81: maybe add that talent development programs are established approaches of talent promotion in a variety of sports with some examples, the focus on table tennis)

• L117: explain, why the study is explorative

Response

The comments have been addressed as following: 

- Excellent adult performance is defined as international level according to the classification levels of Swann (2014) (line 46) with reference to: Swann C, Moran A, Piggott D. Defining elite athletes: Issues in the study of expert performance in sport psychology. Psychology of sport and exercise. 2015;16:3-14.

- The following reference has been added (line 53): Faber IR, Sloot L, Hoogeveen L, Elferink-Gemser MT, Schorer J. Western Approaches for the identification and development of talent in schools and sports contexts from 2009 to 2019-a literature review. High Ability Studies. 2021:1-34.

- This text has been changed into: prospective longitudinal approaches including longitudinal tracking in line with the scoping review of Baker et al. (line 54 and line 73-75).

- More details have been added to clarify the estimate of potential: A fair estimate of an athlete’s potential to reach the international level and the chances to stand on international podia…. (line 59-61)

- The suggested sentence is added (line 63-64) together with the following reference: De Bosscher V, De Knop P, Van Bottenburg M, Shibli S. A conceptual framework for analysing sports policy factors leading to international sporting success. European sport management quarterly. 2006;6(2):185-215.

- Explorative Is deleted; this was mainly due to the small sample, but seems unnecessary.

Methods

• l.144 provide distribution regarding gender and age

• Is there a rule, in which ages the players are nominated for the squads? If I am right, players were between 7 and 11 years old, which seems to be a rather large range for initial talent identification events (if the events were of such type?)

• L.157pp: please double check the presentation of the statistical values. Should read e.g., ICC = 0.91. Please also check formatting (e.g., n should be in italics, see for instance l.144)

• L166-168: sentence seems not to be complete.

• L.170pp: Please explain which “levels” were exactly used (or whether level is the right term for the score that was used). See also general comment.

• L.187: was there also significance testing for normality?

Response

The comments have been addressed as following: 

- The distribution of gender and age is added to the results section (line 219). 

- In table tennis it is common practice that players are selected for more intensive training programs already at an early age, e.g., from 10-12 years or even younger. The context presented within the paper is representative for a common situation in the Netherlands. The first time that players are selected for international competitions (i.e., European Youth Championships) is for the under-15 category. 

- Statistical values were adapted as proposed (line 162-168). 

- Sentence has been reformulated (line 173-175).

- We deleted the word level and changed this into competition performance. The competition rating score is used to reflect the competition performance of a player. More detailed information of the score is added (line 185-194).

- We didn’t perform a statistical test for normality. The normality tests are not very valid for small sample sizes. Therefore, we evaluated normality by comparing mean and median and by visual inspection.

Results:

• L.215: lower values are better performances in speed measures, right? Please note on that, as this is important for the interpretation of the HRs.

• By the way: although I like the idea of referring to the diagnostics elsewhere, it would be helpful to at least provide the units /general shape of the utilized tests) in order to better understand interpretations in this part.

• “best fitting model”: please provide measures for that (and describe the process how it was found in more detail)

• L.226pp: Please state what was used as independent and dependent variables in the regressions (already in the methods section) and add this information also in Table 2.

 

Response

The comments have been addressed as following:

- This information is added as footnote in Table 1. 

- More detailed information about the tests is added to the methods section (line 144-160)

- After considering this formulation, we think it's better to approach this as the final multivariable model instead of the 'best fitting' model. We used a backwards procedure to come to this final model and did not make a comparison. We changed this accordingly. 

- The information about independent and dependent variables is now added to the methods section within the statistical analysis paragraph (line 203-208). 

Discussion

L.250: decimal point instead of comma

L.261: commas before and after “therefore”?

l.290: see general comment on combined approach regarding gender

L.195pp: In line with my comment regarding the introduction of the utilized and investigated measure, I would recommend to extend this paragraph by giving more information why the study “only” focused on a small part of the complex construct of talent and to highlight that this small piece is more a support for coaches and not a replacement (?).

Response

The comments have been addressed as following:

- This is changed as proposed (line 276).

- This is changed as proposed (line 286). 

- Although we indeed agree that an analysis including sex would be the better solution, this is considered inappropriate within this small dataset. Therefore, we only corrected for sex by including sex as a covariate and included this in the limitation paragraph of the discussion section. 

- This information is added to the concluding paragraph (line 320-326). 

Figure 1:

• See general comment. I would recommend deleting ages 0-7 years (not included in the study, and players were potentially not in the system yet. Please also provide Confidence intervals for the survivals (in order to get insight in potential problems due to lower sample sizes for later drop outs). Please add also a label for the columns with n=21 and n=18 (presumably the 2011 and 2012 assessments separately?)

Response

The comments have been addressed as following:

- Ages from 0-7 are deleted as proposed in Fig1. 

- The Kaplan Meier is a descriptive plot to display observed values. For this reason, confidence intervals cannot be calculated. Because they are observed values, you have no CI.

- Column labels have been added to Table 1.

Once again, thank you for your feedback that helped us to improve our article.

---

## [Decision Letter · Decision Letter 1]

20 Dec 2022

PONE-D-22-23471R1Can perceptuo-motor skills outcomes predict future competition participation/drop-out and competition performance in youth table tennis players? A 9-year follow-up studyPLOS ONE

Dear Dr. Faber,

Thank you for submitting your manuscript to PLOS ONE. After careful consideration, we feel that it has merit but does not fully meet PLOS ONE’s publication criteria as it currently stands. Therefore, we invite you to submit a revised version of the manuscript that addresses the points raised during the review process.

ACADEMIC EDITOR: Dear authors,

The authors improved their work when compared with the previous version. Reviewer 1 already recommended acception of the work while reviewer 2 suggested minor revision. This is also my opinion because there are still some possibilities to developt the content of the manuscript. Please check all comments made by reviewer 2 and address.

BR 

We look forward to receiving your revised manuscript.

Kind regards,

Rafael Franco Soares Oliveira

Academic Editor

PLOS ONE

Journal Requirements:

Additional Editor Comments:

Dear authors,

The authors improved their work when compared with the previous version. Reviewer 1 already recommended acception of the work while reviewer 2 suggested minor revision. This is also my opinion because there are still some possibilities to developt the content of the manuscript. Please check all comments made by reviewer 2 and address all of them. I believe that if all is satisfied, the paper will be accepted.

BR

Reviewers' comments:

Reviewer's Responses to Questions

**Comments to the Author**

1. If the authors have adequately addressed your comments raised in a previous round of review and you feel that this manuscript is now acceptable for publication, you may indicate that here to bypass the “Comments to the Author” section, enter your conflict of interest statement in the “Confidential to Editor” section, and submit your "Accept" recommendation.

Reviewer #1: All comments have been addressed

Reviewer #2: (No Response)

2. Is the manuscript technically sound, and do the data support the conclusions?

Reviewer #1: Yes

Reviewer #2: Yes

3. Has the statistical analysis been performed appropriately and rigorously? 

Reviewer #1: Yes

Reviewer #2: Yes

4. Have the authors made all data underlying the findings in their manuscript fully available?

Reviewer #1: Yes

Reviewer #2: No

5. Is the manuscript presented in an intelligible fashion and written in standard English?

Reviewer #1: Yes

Reviewer #2: Yes

6. Review Comments to the Author

Reviewer #1: First of all, the reviewer would like to thank the authors for their work and efforts in trying to improve sports science knowledge

Reviewer #2: Reviewer Comments:

Can perceptuo-motor skills outcomes predict future competition participation/drop-out and competition performance in youth table tennis players? A 9-year follow-up study: PONE-D-22-23471_R1

Thank you for submitting your revised manuscript to the PLoS One. From my point of view, the authors should be complimented for their revisions made on the paper. Most of the reviewers’ comments were addressed in a satisfying manner. However, there are still some issues that are listed below and should be considered (lines refer to the revised manuscript with track changes):

1) While I appreciate structuring the paper according to your two RQs, this is still missing in the abstract. Consider introducing RQ1 and RQ2 here as well. In doing so, the results within the abstract can me assigned to that structure easily. This would enhance understanding.

2) L.99: Although you almost entirely adjusted the naming of the criterion variable for RQ2 (competition performance), there are still some situation were you refer to level. Consider removing level here as well as in l.186 if appropriate. You are referring to a continuous variable in your analyses (performance score). Therefore, avoid using “level” which might suggest a nominal outcome. By the way: conder using “future competition performance” here.

3) L.104 and throughout: check formatting of statistical values again. R^2 in italics?

4) L.187pp: while I appreciate your elaborations on the score, it would be helpful for the readership to provide a formula for this score, if possible. This would also help to get a better impression of your multilevel result…. If not just leave it as it is.

5) L.202 pp: Thank you for including information on KMC here. While this represents an interesting analysis, this- in the current version- does not refer to RQ1 or RQ2. Those RQs solely refer to the predictive value of the performed perceptuo motor test. Therefore I would suggest either including this analysis and their result into a RQ- presumably this could be RQ1a and the results referring to the tests could be RQ1b, or removing this analyses from the manuscript.

6) L.210: provide more rationale why multi- and univariate approaches were followed a potential reason could be investigating single tests power (univariate), but also the test batterys power (multivariate). Just shortly elaborate on that, please.

7) L.220: were these indeed different talent days for boys and girls? Or are the number 21 and 18 incidentally the same as those for the gender distribution? Please clarify. By the way: if there were different talent days, there is still further reason to discriminate boys and girls in your analyses: 1 year difference of testing dates, level of participants comparable among boys and girls?, …. Maybe at least add some thoughts to the discussion.

8) Table1: Please provide results for gender and age as covariates, please (as a follow-up in your answers to the latter revision). It is important to know, whether the covariates were significantly influencing the results, or not. Please state and also present as text in the results section. If there is a significant influence, you should also discuss whether there could be reasoned assumptions as to whether the prognostic relevance among genders could be different? [which would be maybe reflected by an interaction effect!?.

9) Table 1: Please provide at least 1 decimal for the Means of your test (in line with the given SDs that are already presented with 1 decimal.

10) L.248: This subheading does not fit to the subheading of RQ1. Please use the same “structure”: should this also read “Predicting …” ?

11) L.250: delete “(p <.05”) as you present each exact p afterwards.

12) L.252: (Oxford) comma before “and eye-hand coordination!?

13) Table 2: Can you also present the R^2s for each model in the Table? This would be kind of an effect size and, thus, in line with all further analyses where you present effect sizes (e.g., HR).

14) L.317pp: As already mentioned, I appreciate presenting the limitations regarding sex and age. However, this is a severe issue in your analyses. Therefore, I recommend extending this section: what are your expectations about concrete limitations/restrictions of generalizability of your analyses towards the predictive value of the conducted tests. For me, it is by no means clear that the predictive power of such tests is the same for boys and girls. And by the way, this is well-known problem in talent research. Already William & Reilly (2000) claim for the consideration and comparison of both genders when investigating the predictive value of talent predictors. Just a thought: I know, that your sample size is too low for performing separate analyses (and getting significant results at the same time). However, did you perform those analyses also separately and compared e.g. the regression coefficients of those (non-significant) models with those given for the total sample. This (together with the results of the covariates) could indicate as to whether the differences between gender could be more or less neglected in your study. I would appreciate reading about such points here in this part of the discussion.

Thank you again for this interesting paper. I hope that my comments are appreciated and help to further enhance the quality of the paper.

7. PLOS authors have the option to publish the peer review history of their article (what does this mean?). If published, this will include your full peer review and any attached files.

Reviewer #1: No

Reviewer #2: No

---

## [Author Response · Author response to Decision Letter 1]

17 Jan 2023

Point-by-point response

Journal Requirements:

Response

We checked the references. No updates were necessary for this revision. 

Additional Editor Comments:

Dear authors,

The authors improved their work when compared with the previous version. Reviewer 1 already recommended acception of the work while reviewer 2 suggested minor revision. This is also my opinion because there are still some possibilities to developt the content of the manuscript. Please check all comments made by reviewer 2 and address all of them. I believe that if all is satisfied, the paper will be accepted.

Response

Thank you for your positive response regarding our paper.

Reviewers' comments:

Reviewer #1: 

First of all, the reviewer would like to thank the authors for their work and efforts in trying to improve sports science knowledge. 

Response

Thank you for your positive response regarding our paper and accepting it for publication. 

Reviewer #2: 

Thank you for submitting your revised manuscript to the PLoS One. From my point of view, the authors should be complimented for their revisions made on the paper. Most of the reviewers’ comments were addressed in a satisfying manner. However, there are still some issues that are listed below and should be considered (lines refer to the revised manuscript with track changes):

Response

Thank you for your positive response and critical feedback that helps us to improve our paper. Lines refer to the version with track changes. 

1) While I appreciate structuring the paper according to your two RQs, this is still missing in the abstract. Consider introducing RQ1 and RQ2 here as well. In doing so, the results within the abstract can me assigned to that structure easily. This would enhance understanding.

Response

References to RQ1 and RQ2 are now included within the abstract. 

2) L.99: Although you almost entirely adjusted the naming of the criterion variable for RQ2 (competition performance), there are still some situation were you refer to level. Consider removing level here as well as in l.186 if appropriate. You are referring to a continuous variable in your analyses (performance score). Therefore, avoid using “level” which might suggest a nominal outcome. By the way: conder using “future competition performance” here.

Response

Thank you for your suggestions. We deleted 'level' at line 99, line 186 and line 298.

3) L.104 and throughout: check formatting of statistical values again. R^2 in italics?

Response

This has been changed as proposed. 

4) L.187pp: while I appreciate your elaborations on the score, it would be helpful for the readership to provide a formula for this score, if possible. This would also help to get a better impression of your multilevel result…. If not just leave it as it is.

Response

Unfortunately, it is not possible to provide an easy calculation for the performance score. However, the complete description how the rating score is calculated can be found here: De Elo-ranglijsten toelichting en berekeningswijze (nttb-ranglijsten.nl). This webpage is in Dutch and provides a comprehensive outline. The link is added in the method section (line 190). 

5) L.202 pp: Thank you for including information on KMC here. While this represents an interesting analysis, this- in the current version- does not refer to RQ1 or RQ2. Those RQs solely refer to the predictive value of the performed perceptuo motor test. Therefore I would suggest either including this analysis and their result into a RQ- presumably this could be RQ1a and the results referring to the tests could be RQ1b, or removing this analyses from the manuscript.

Response

We decided to delete the KMC in the revised version to focus directly on the proposed research questions. 

6) L.210: provide more rationale why multi- and univariate approaches were followed a potential reason could be investigating single tests power (univariate), but also the test batterys power (multivariate). Just shortly elaborate on that, please.

Response

This information is added to the statistical analysis as proposed (line 211-214). 

7) L.220: were these indeed different talent days for boys and girls? Or are the number 21 and 18 incidentally the same as those for the gender distribution? Please clarify. By the way: if there were different talent days, there is still further reason to discriminate boys and girls in your analyses: 1 year difference of testing dates, level of participants comparable among boys and girls?, …. Maybe at least add some thoughts to the discussion.

Response

The talent days included both boys and girls. Indeed the numbers are the same by coincidence. The exact numbers of boys and girls per year are added to the results section (line 223-224) to clarify this. 

8) Table1: Please provide results for gender and age as covariates, please (as a follow-up in your answers to the latter revision). It is important to know, whether the covariates were significantly influencing the results, or not. Please state and also present as text in the results section. If there is a significant influence, you should also discuss whether there could be reasoned assumptions as to whether the prognostic relevance among genders could be different? [which would be maybe reflected by an interaction effect!?.

Response

Information about the covariates' significance in the univariable and multivariable models for both RQ1 and RQ2 is added in table 1 and 2 and in the text of the results section (line 243-245 and line 258-262). Based on these outcomes it appears that test age and sex were significant covariates within some of the univariable Cox regression analyses, but not in the final multivariable one. Influences of test age and sex are further discussed in the discussion section (line 300-305, line 327-332, line 337-341). 

9) Table 1: Please provide at least 1 decimal for the Means of your test (in line with the given SDs that are already presented with 1 decimal.

Response

Decimals are now provided in Table 1 as proposed. These changes are not shown with track changes to preserve readability. 

10) L.248: This subheading does not fit to the subheading of RQ1. Please use the same “structure”: should this also read “Predicting …” ?

Response

This has been changed as proposed. 

11) L.250: delete “(p <.05”) as you present each exact p afterwards.

Response

This has been changed as proposed. 

12) L.252: (Oxford) comma before “and eye-hand coordination!?

Response

This has been changed as proposed. 

13) Table 2: Can you also present the R^2s for each model in the Table? This would be kind of an effect size and, thus, in line with all further analyses where you present effect sizes (e.g., HR).

Response

R2 have been added to table 2 as proposed.

14) L.317pp: As already mentioned, I appreciate presenting the limitations regarding sex and age. However, this is a severe issue in your analyses. Therefore, I recommend extending this section: what are your expectations about concrete limitations/restrictions of generalizability of your analyses towards the predictive value of the conducted tests. For me, it is by no means clear that the predictive power of such tests is the same for boys and girls. And by the way, this is well-known problem in talent research. Already William & Reilly (2000) claim for the consideration and comparison of both genders when investigating the predictive value of talent predictors. Just a thought: I know, that your sample size is too low for performing separate analyses (and getting significant results at the same time). However, did you perform those analyses also separately and compared e.g. the regression coefficients of those (non-significant) models with those given for the total sample. This (together with the results of the covariates) could indicate as to whether the differences between gender could be more or less neglected in your study. I would appreciate reading about such points here in this part of the discussion.

Response

As proposed, information about the covariates' significance in the univariable and multivariable models for both RQ1 and RQ2 is added in table 1 and 2 and in the text of the results section (line 243-245 and line 258-262). Based on these outcomes it appears that test age and sex were significant covariates within some of the univariable Cox regression analyses, but not in the final multivariable one. For the prediction of competition performance, only test age appeared a significant covariate. The issue of possible (interaction) effects of sex is pointed out in the discussion part (line 300-305, line 327-332, line 337-341). 

Thank you again for this interesting paper. I hope that my comments are appreciated and help to further enhance the quality of the paper.

Response

Thank you for your feedback, this is much appreciated.

---

## [Decision Letter · Decision Letter 2]

31 Jan 2023

Can perceptuo-motor skills outcomes predict future competition participation/drop-out and competition performance in youth table tennis players? A 9-year follow-up study

PONE-D-22-23471R2

Dear Dr. Dr. Irene R. Faber,

We’re pleased to inform you that your manuscript has been judged scientifically suitable for publication and will be formally accepted for publication once it meets all outstanding technical requirements.

Kind regards,

Rafael Franco Soares Oliveira

Academic Editor

PLOS ONE

Additional Editor Comments (optional):

Congratulations! Our recommendation is to accept your manuscript!

Reviewers' comments:

Reviewer's Responses to Questions

**Comments to the Author**

1. If the authors have adequately addressed your comments raised in a previous round of review and you feel that this manuscript is now acceptable for publication, you may indicate that here to bypass the “Comments to the Author” section, enter your conflict of interest statement in the “Confidential to Editor” section, and submit your "Accept" recommendation.

Reviewer #2: All comments have been addressed

2. Is the manuscript technically sound, and do the data support the conclusions?

Reviewer #2: Yes

3. Has the statistical analysis been performed appropriately and rigorously? 

Reviewer #2: Yes

4. Have the authors made all data underlying the findings in their manuscript fully available?

Reviewer #2: (No Response)

5. Is the manuscript presented in an intelligible fashion and written in standard English?

Reviewer #2: Yes

6. Review Comments to the Author

Reviewer #2: Thank you for the revised version of your paper. All comments have been adressed accordingly and the authors should be complimented for their work! I have no further comments.

7. PLOS authors have the option to publish the peer review history of their article (what does this mean?). If published, this will include your full peer review and any attached files.

Reviewer #2: No

---

## [Editor Report · Acceptance letter]

1 Feb 2023

PONE-D-22-23471R2 

Can perceptuo-motor skills outcomes predict future competition participation/drop-out and competition performance in youth table tennis players? A 9-year follow-up study 

Dear Dr. Faber:

I'm pleased to inform you that your manuscript has been deemed suitable for publication in PLOS ONE. Congratulations! Your manuscript is now with our production department. 

Kind regards, 

on behalf of

Dr. Rafael Franco Soares Oliveira 

Academic Editor

PLOS ONE